# Reasons for Regularly Using Heated Tobacco Products among Adult Current and Former Smokers in Japan: Finding from 2018 ITC Japan Survey

**DOI:** 10.3390/ijerph17218030

**Published:** 2020-10-31

**Authors:** Steve S. Xu, Gang Meng, Mi Yan, Shannon Gravely, Anne C. K. Quah, Janine Ouimet, Richard J. O’Connor, Edward Sutanto, Itsuro Yoshimi, Yumiko Mochizuki, Takahiro Tabuchi, Geoffrey T. Fong

**Affiliations:** 1Department of Psychology, University of Waterloo, Waterloo, ON N2L 3G1, Canada; gmeng@uwaterloo.ca (G.M.); mi.yan@uwaterloo.ca (M.Y.); shannon.gravely@uwaterloo.ca (S.G.); ackquah@uwaterloo.ca (A.C.K.Q.); j2ouimet@uwaterloo.ca (J.O.); gfong@uwaterloo.ca (G.T.F.); 2Division of Cancer Prevention and Population Sciences, Department of Health Behaviors Roswell Park Comprehensive Cancer Center, Buffalo, NY 14263, USAedward_sutanto@live.com (E.S.); 3Division of Tobacco Policy Research, National Cancer Center Japan, 5-1-1 Tsukiji, Chuo-ku, Tokyo 104-0045, Japan; iyoshimi@ncc.go.jp; 4Japan Cancer Society, 13th Floor, Yurakucho Center Bldg. 2-5-1, Yurakucho, Chiyoda-ku, Tokyo 100-0006, Japan; yumiko.mochizuki@gmail.com; 5Cancer Control Center, Osaka International Cancer Institute, Chome-1-69 Otemae, Chuo Ward, Osaka 541-8567, Japan; tabuchitak@gmail.com; 6School of Public Health and Health Systems, University of Waterloo, Waterloo, ON N2L 3G1, Canada; 7Ontario Institute for Cancer Research, Toronto, ON M5G 0A3, Canada

**Keywords:** heated tobacco products, cigarettes, Japan, reasons, current smoker, former smoker

## Abstract

The market growth of heated tobacco products (HTPs), such as IQOS, Ploom TECH, and glo, has increased dramatically in Japan since 2016. Little is known about the reasons why current and former smokers are using HTPs. The data for this cross-sectional study were from the 2018 (Wave 1) International Tobacco Control (ITC) Japan Survey, a national web-based survey of 4500 people, including 658 current HTP users, of whom 549 were concurrently smoking cigarettes and 109 were former smokers. The most common reasons for regularly using HTPs were: beliefs that HTP are less harmful than cigarettes to themselves (90.6%) or to others (86.7%), enjoyment (76.5%), and social acceptability (74.4%). About half of current smokers (55.1%) reported using HTPs because these products might help them quit smoking. However, a near-equal percentage (52.0%) of current smokers reported using HTPs to replace some of the cigarettes they smoked so that they did not have to give up smoking altogether. If smokers are using HTPs to complement rather than quit their smoking, then the harm reduction potential of HTPs suggested by the toxicity studies will be diminished.

## 1. Introduction

HTPs heat tobacco in loose leaf form or contained in tobacco sticks, plugs, or capsules using a battery-powered heating system. Tobacco is heated (not combusted, as in traditional cigarettes) to generate an inhalable nicotine-containing aerosol [1]. Studies of HTP cytotoxicity and emissions of nicotine and toxic nitrosamines have found that HTPs are lower in both compared to cigarettes, suggesting that it may be beneficial for smokers to completely switch to an HTP [2,3,4]. In May 2020, the Food and Drug Administration (FDA) in the Unites States granted limited authorization to market IQOS as a modified risk tobacco product [5], allowing claims that IQOS reduces exposure to harmful chemicals, but not allowing claims that IQOS reduces harm.

Beginning in 2014, Philip Morris International (PMI) launched IQOS initially as a test market in Nagoya, then nationwide by 2016; Japan Tobacco (JT) launched Ploom TECH in March 2016; and British American Tobacco (BAT) launched glo in December 2016. Japan represents a strategic test market for HTPs because: (1) adult smoking prevalence was relatively high (19.3% in 2016) [6]; (2) Japan has a tobacco-friendly business environment in which the government owns one-third of JT, and tobacco control policies are weaker compared to other high-income countries [7]; and (3) the sale of nicotine-containing e-cigarettes is banned, thus, there is little competition in the alternative nicotine delivery products market [8].

Accompanying their respective HTP launches, tobacco companies have carried out intensive marketing campaigns to promote HTPs as “less harmful” and “cleaner” alternatives to combustible cigarettes [6,9,10]. HTP advertisements typically include messages such as “producing less than 1% of the odor” or “more than 99% reduction in the levels of measured constituents compared to cigarette smoke” [11]. Recently, tobacco companies have promoted HTPs as being a better alternative to combustible cigarettes with external benefits of being more socially acceptable to others, and more conveniently used in public places where smoking is prohibited [12]. Aggressive marketing efforts to normalize HTP use have resulted in a surge of HTP initiations, especially among current or former cigarette smokers [8]. A nationally representative online survey of Japanese adults reported that the prevalence of IQOS and Ploom TECH use were both 0.3% in 2015, and by 2017, the prevalence of HTP use had increased to 3.6%, with IQOS being the leading brand (IQOS: 3.6%, Ploom TECH: 1.2%, and glo: 0.8%) [4]. Another indication of the growing popularity of HTPs is their rising market share, which reached 21.7% in total tobacco sales volume by September 2018 [13].

Two independent qualitative studies have reported reasons why consumers use HTPs. In a focus group study conducted in Japan and Switzerland, IQOS users cited attractive packaging; lack of ash, smell, and smoke; and greater acceptability of HTPs compared to cigarettes [14]. Similarly, an in-depth one-to-one interviews of 30 current and ex-IQOS users in London, United Kingdom, respondents reported several reasons for use, including: believing IQOS was a healthier alternative to smoking, for some as a pathway to quit smoking, the novelty of the product, appealing packaging, and to use in places where they could not smoke [15].

Given that tobacco companies market HTPs to consumers as a better alternative to cigarette smoking, there is currently a lack of independent research of large national samples of current and former smokers examining the reasons for HTP use, particularly in Japan, which is the largest global market with no official advertising restrictions for HTPs. Current or potential HTP users thus may not understand the implications of not completely switching from cigarettes to HTPs (e.g., in order to reduce toxicity from continued smoking). The objectives of the present descriptive study were to explore the reason(s) why HTP users are using them regularly among a large national sample of Japanese smokers and former smokers.

## 2. Methods 

Data were from Wave 1 of the International Tobacco Control Japan Survey (ITC JP1), a web-based survey of a national sample of 4615 adult (aged 20+) exclusive cigarette smokers (who smoked at least monthly), exclusive HTP users (who used HTPs at least weekly), concurrent users (dual users who used both HTPs and cigarettes), and non-users (who did not smoke or use HTPs). The sample was recruited from Rakuten Insight’s proprietary online panel and the survey was conducted from 3 February to 2 March 2018, with quotas for region of residence, gender, and age, to ensure that the final sample was proportional to stratum sizes from Japan census data. The survey response rate was 45.1%, and the cooperation rate was 96.3%. A detailed description of the ITC Japan Survey sample and methods is reported elsewhere [16,17]. Study procedures and materials were reviewed and cleared by the Office of Research Ethics, University of Waterloo, Canada (ORE# 22508/31428).

### 2.1. Study Sample 

The current study included 549 concurrent users (smokers who also use an HTP at least weekly), and 109 former smokers (quit smoking cigarettes completely, and use an HTP at least weekly). Of 109 former smokers, 90 stopped smoking 2 years ago or less, and 19 more than 2 years ago. A total of 55 exclusive HTP users who were occasional smokers (*n* = 47, currently smoke cigarettes less than monthly) or never smokers (*n* = 88) were excluded from the study. We excluded these occasional smokers because it was not clear whether their infrequent smoking was indicative of being a current or former smoker. By excluding such respondents, we ensured that the dataset analyzed included respondents whose user group was clearly identified.

### 2.2. Measures

#### 2.2.1. Independent Variables

##### Smoking Status

Respondents reported their cigarette smoking frequency at the time of the survey, and were categorized as a ‘smoker’ if they smoked at least monthly or as a ‘former smoker’ if they reported having quit smoking completely.

##### HTP Use

All respondents were asked if they used an HTP. If they used HTP regularly (at least weekly) then they were included in the study.

##### Concurrent User (Current Smokers) vs. Exclusive HTP User (Former Smoker)

Based on both smoking and HTP use status, respondents were further classified as a concurrent user (smoker also uses HTP) or as an exclusive HTP user (former smoker uses HTP).

Covariates included gender (female versus (vs.) male) and age, which were collected by the survey firm at the time of recruitment, and verified at the time of the survey.

#### 2.2.2. Outcome: Reasons for HTP Use

Reasons for HTP use among current and former smokers were evaluated using the following questions: *“Which of the following are reasons that you use heated tobacco products?”.* Respondents had the option of selecting from 20 possible reasons for their HTP use. These reasons were then categorized into eight broad themes, which in part were derived from the study by Patel et al. that categorized responses about reasons for e-cigarette use, among a national sample of adults in the US [18]: health-related factors (e.g., harm reduction), convenience, and (social) consideration. For current smokers, we provided three additional themes for HTP use were: to help cut down smoking, to help stop smoking, and to help stay smoking (e.g., they can use an HTP in places where they cannot smoke) (Table 1). Other reasons were asked, but they were not included in the present study because they were reasons for initiating HTPs rather than for regularly using HTPs (curiosity, someone offered me one, a health professional advised that I switch to HTPs).

For each reason, the response options were “yes”, “no”, “refused”, or “do not know”. Respondents could select multiple reasons for using HTPs; therefore, these reasons were not mutually exclusive. For analysis purposes, “refused” or “do not know” responses were excluded. Response rates for each item (“yes” or “no”) ranged from 98.6% for “family or friends use heat-not-burn products” to 66.7% for “experts, like doctors and scientists, use them”. The average response rate for completing the items was 88.6%. A validation test on variance estimates indicated that the missing values are missing at random.

#### 2.2.3. Data Analysis

Unweighted frequencies were used to describe the study sample (Table 2). All other analyses were conducted on weighted data. In brief, a raking algorithm was used to calibrate the cross-sectional weights on smoking status, HTP-use, geographic region, and demographic measures to adjust for potential disproportional sampling of sub-groups among these categories (such as the oversampling of the dual user group) to make respondents within each of the sub-groups representative of the corresponding population. The weight calibration was done using benchmarks from the 2017 Japan Society and Tobacco Internet Study (JASTIS). Further details about weighting can be found in the ITC Japan Wave 1 Technical Report [16].

The predicted marginal standardization method (PREDMARG) using logistic regression models in SAS callable SUDAAN v11. (SAS Institute Inc. 2013, Cary, NC, USA) was used for estimating prevalence [19]. Covariates included age, gender, tobacco use status (categorized into current vs. former smokers), and HTP use frequency (categorized into daily vs. weekly). The model did not control for education and income because they were not significant in the initial bivariate analyses.

Analyses were first conducted by combining current and former smokers into one model to examine overall reasons for using HTPs prevalence estimates, and then were separated into current and former smokers to examine their potential differences. General linear contrasts of the predicted marginals in the corresponding models were specified for significance testing of percent differences between current smokers and former smokers. All confidence intervals (CIs) and statistical significance were tested at the 95% confidence level.

## 3. Results

### 3.1. Sample Characteristics

Table 2 presents the unweighted sample characteristics of current and former smokers using HTPs in the ITC JP1 Survey. The majority were men (63.7%), current smokers using HTPs or concurrent users (83.4%), and daily cigarette smokers (79.6%). The average age was 41.4. By age group, those aged 40+ were the largest group (49.5%), followed by those aged 30–39 (31.3%), and aged 20–29 (19.2%).

### 3.2. Overall Reasons for Using HTPs

Table 3 presents the reasons for using HTPs among current and former smokers in order of their frequency. The top five reasons for using HTPs among both current and former smokers were the belief that HTPs are less harmful to either themselves (90.6%) or others (86.7%), personal enjoyment (76.5%), and that HTPs are more acceptable to others (74.4%). In addition, 61.8% of HTP users believed that HTPs could reduce stress, 55.9% used HTPs because family or friends used them, and 51.8% reported that HTPs gave them something to do or occupy their time. In contrast, few respondents used HTPs for the following reasons: HTPs are more affordable than cigarettes (20.1%), HTPs have attractive packaging (16.1%), to control their appetite or weight (16.2%), to look cool (12.5%), and because experts such as doctors and scientists use HTPs (10.3%).

### 3.3. Reasons for Using HTPs among Current Smokers

Among current smokers, the top five reasons for using HTPs were the belief that HTPs are less harmful to either themselves than smoking (88.0%) or others (83.9%), personal enjoyment (75.2%); more acceptable to others (72.5%), and that family or friends use them (58.9%). About or over half reported the personal needs of stress reduction (52.8%), giving them something to do or occupying their time (54.1%), for convenience in places where smoking cigarettes is banned (49.4%). Meanwhile, 64.4% of current smokers reported using HTPs to reduce their cigarette consumption. Additionally, 55.1% current smokers reported using HTPs because HTPs might help them quit smoking; however, 52.0% reported using HTPs to replace some of their cigarettes so that they do not have to give up smoking cigarettes altogether.

### 3.4. Reasons for Using HTPs among Former Smokers

Among former smokers, the top five reasons include HTPs being less harmful to themselves (96.6%) or to others (92.3%), personal enjoyment (78.8%), HTPs being more acceptable to others (77.7%), and stress reduction (76.7%). About or over half of former smokers reported using HTPs because they taste good (58.6%), and that family or friends use HTPs (50.9%). Only 21.5% cited saving money as a reason.

### 3.5. Difference in the Reasons for Using HTPs between Current and Former Smokers

Compared to current smokers, former smokers were more likely to cite the reasons: HTPs are as less harmful to themselves (95.8% vs. 88.0%, *p* < 0.01) or people around them (91.4% vs. 83.8%, *p* < 0.05), for reducing stress (76.7% vs. 52.8%, *p* < 0.001), and have good taste (58.6% vs. 37.5%, *p* < 0.001). In contrast, current smokers were more likely to report that they could use HTPs in places where smoking cigarettes are banned (49.4% vs. 26.2%, *p* = 0.02), to make socialising easier (40.0% vs. 27.3%, *p* < 0.05), and because of the attractiveness of the heating/charging device (33.4% vs. 17.5%, *p* < 0.01) or packaging (22.3% vs. 4.0%, *p* < 0.001).

## 4. Discussion

Aside from industry-funded research, little is known about the reasons for HTP use in Japan. This study quantitatively explored reasons for HTP use among a representative sample of Japanese current and former smokers. Our findings show that the most common reason cited is that HTPs are less harmful than combustible cigarettes to themselves and others. This finding is consistent with data showing that almost two thirds of HTP users in Japan perceived HTPs to be less harmful than combustible cigarettes [20]. A similar finding of an online study in Korea shows that current IQOS users reported using IQOS because they perceived IQOS to be less harmful than combustible cigarettes [21]. Harm reduction-related reasons for HTP use appears to be consistent with the marketing messages of major tobacco companies in Japan [10,11,22].

Second, consideration for others emerged as a common motivator for using HTPs. Over two thirds of respondents considered HTP use to be more socially acceptable than smoking cigarettes. Another study found that characterizing HTPs as ‘clean’ aligns well with the Japanese cultural value of respecting others [14]. JT’s survey of over 3000 Japanese consumers reported that many used HTPs out of consideration of others, and acceptability to others [11]. Being able to use HTPs in places where smoking cigarettes is inappropriate or banned was also a major motivator. Almost half of adults cited opting for HTPs so they could use them in places where smoking cigarettes is banned. Indeed, a substantial proportion of HTP use within indoor public spaces in Japan has been reported by an ITC study [23]. Tobacco companies have been marketing the benefits of using HTPs for these situations, which may have the potential to normalize HTP use in Japanese society [11].

Third, Japanese cultural values include respecting the opinion of others. Our finding confirms that family or friends using HTPs was a common reason for using HTP. Social influencers played a role in initiating HTP use, as one third of respondents reported that people in the media or other public figures led them to use HTPs, which confirms Tabuchi et al.’s 2019 finding [8]. About one third of respondents indicated that they used HTPs because someone else offered them. While giving and gifting cigarettes has historically been a widespread tradition in East Asia [24], further study may be needed to investigate whether this is also the case for HTPs.

Fourth, at the time of the survey, the market share of premium-brand cigarettes (retail price 470 yen or above, 20 sticks/pack) was 16.7%, that of mid-priced-brand cigarettes (400–460 yen/pack) was 76.9%, and that of economy-brand cigarettes (below 400 yen/pack) was 6.4% [25]. The market share of IQOS heatsticks (retail price 460 yen/pack) was 74.9%, BAT’s neosticks (420 yen/pack) was 17% and JT’s Ploom TECH tobacco refills (460 yen/pack) was 8.1% [26]. The fact that 85% of HTPs sold in Japan were priced right below the price of premium brand cigarettes suggested that tobacco companies did not intend to attract smokers to take up HTPs with an affordable price. Our finding is consistent with the HTP pricing strategy, in that most respondents did not report initiating HTP use in order to save money. Replacing premium brand cigarettes with the most popular HTP—IQOS heatsticks—may not incur significant cost saving, and replacing the most popular mid-priced brand cigarettes with IQOS heatsticks could cost more if the tobacco consumption pattern remains the same. JT’s 2018 study of Ploom TECH users showed the total tobacco consumption of dual users of HTPs and cigarette actually increased, thus possibly incurring more cost [27].

Both independent and industry-sponsored studies confirmed that most HTP users concurrently smoke combustible cigarettes [9,21,27,28]. While it is feasible that some dual users may be in the midst of a quit attempt, others may be using HTPs to maintain their cigarette-smoking habit at times or places that they cannot smoke [29]. The literature of PMI-sponsored research has emphasized this quit-smoking pathway, claiming that around 70% of IQOS users have completely or predominantly switched from smoking combustible cigarettes to “smoke-free” products, such as HTPs. This result, in return, has been used to support its projection that switching to HTPs could save the lives of millions of current smokers [6]. Our study shows that two-thirds of dual users reported that HTPs help them reduce their cigarette consumption. The reduction however does not mean all smokers intend to replace combustible cigarettes with HTPs completely. Our finding reveals that half actually intend to use HTPs to help staying smoking, suggesting that complementing smoking is at least an equally important pathway of using HTPs among current smokers in Japan. Although switching completely from cigarettes to HTPs may be associated with positive public health outcomes, sustained dual use would not have that same benefit, and indeed it is unclear at this point whether it would have any positive public health effect.

Our results demonstrate that current and former smokers cited different reasons for using HTPs. Generally, former smokers were more likely to report internal and self-oriented, reasons, such as “less harmful” to themselves and others, stress reduction, and good taste. In contrast, current smokers were more likely to be motivated by external and social factors such as using them in places where smoking cigarettes is prohibited, product attractiveness (e.g., they perceive that HTPs are more attractive than cigarettes), and because it makes socializing easier. It also appears that HTP marketing messages have been shifting to more frequently including these external factors which are more geared towards dual use, rather than just demonstrating the internal benefits that mainly target those who want to quit cigarette smoking with the help of HTPs [11,12,13]. Therefore, it appears that the tobacco industry is employing varying marketing strategies that will appeal to different types of users. Further studies are needed to examine if the shift of HTP marketing discourse has led to more smokers taking up HTPs for these alternative reasons suggested by newer marketing messages.

We compared the reasons for current smokers using HTPs in Japan with a recent ITC study of the reasons for current smokers using nicotine vaping products (NVPs; e.g., e-cigarettes) in the 2018 ITC Four Country Smoking and Vaping Survey, conducted in the United States, Canada, England, and Australia [30]. Some of the most common reasons for using NVPs were similar to the reasons for current smokers using HTPs in the present study, including the belief that NVPs were “less harmful to me”: 74.3% among male and 74.5% among female concurrent smokers compared to 88% of current HTP–cigarette dual users in Japan, and that vaping was less harmful to others: 69.8% among male and 77.8% among female concurrent NVP–cigarette users compared to 83.8% among concurrent HTP–cigarette users in Japan. The higher social acceptability of NVPs was another highly prevalent reason for vaping (69.8%/73.4%) that was very similar to the prevalence of this reason for using HTPs (72.5%).

There were, however, two notable differences in the reasons for using NVPs vs. the reasons for using HTPs. First, a substantially higher percentage of concurrent NVP-cigarette users stated that they used NVPs because they could use them in places where smoking cigarettes is banned (63.1% among males, 64.3% among females) vs. concurrent HTP–cigarette users in Japan (49.4%). This may be due to the presence of strong and enforced smoke-free laws in those four countries, which would highlight the perceived benefits of vaping, relative to the weak and weakly enforced smoke-free laws in Japan [31]. Second, concurrent NVP–cigarette users were much more likely to cite saving money as a reason for using NVPs (64.7% among males, 64.9% among females) vs. concurrent HTP–cigarette users in Japan (20.3%), reflecting both the relatively lower cost of vaping vs. smoking in the four countries and the nearly equal cost of using HTPs vs. smoking in Japan and elsewhere [30], and the overall marketing and positioning of HTPs as a high-end product [28,32].

Finally, with respect to those reasons related to quitting or reducing smoking, concurrent HTP–cigarette users in Japan were less likely to report using HTPs to stop smoking (55.1%) than concurrent NVP–cigarette users in the four countries (68.7% of males, 73.7% of females). Similarly, HTP users in this study were also less likely to report using HTPs to cut down on smoking (64.4%) than NVP–cigarette concurrent users (77.1% of males, 85.3% of females). The lower linkage between HTP use and quitting/reducing smoking may reflect a lower percentage of Japanese smokers using HTP to quit smoking and/or the lack of messaging about the possible health benefits of HTPs, either from the industry or from health authorities.

Strengths of our study include using of a large national population sample of current and former smokers to examine the reasons for HTP use; and providing a wide variety of options for reason of HTP use, covering eight broad themes: harm reduction, convenience, social consideration, product attractiveness, personal benefits, help cut down smoking, help stop smoking, and help stay smoking. There are, however, several limitations in this study. First, as this is a cross-sectional study, temporality and causal information between the reasons and tobacco use behavior are not available. Second, we did not ask former smokers if they had used HTP to help them to either completely quit or stay abstinent from cigarette smoking. Further research with additional waves of the ITC Japan Survey will evaluate whether respondents’ cited reasons of using HTPs will impact future tobacco use behavior.

## 5. Conclusions

Our findings demonstrate that perceived health benefits, along with accompanying social and personal benefits, are the main reasons for using HTPs regularly among current and former smokers. These positive perceptions of HTP use point to the success of tobacco industry’s effort to normalize HTP use among consumers. Given that more than half of current smokers in Japan intend to use HTPs to help stay smoking cigarettes, this may reduce, or eliminate altogether, the possible public health benefits of HTPs suggested by the toxicity studies.

## Figures and Tables

**Table 1 ijerph-17-08030-t001:** Reasons for using heated tobacco products (HTPs) among current and former smokers.

Theme	Reason	User Group Asked
Harm reduction	Less harmful to my health than ordinary cigarettes	All respondents:Current smokers and former smokers
	Less harmful to the health of people around me than ordinary cigarettes	
Convenience	Use them in places where smoking ordinary cigarettes is banned	
Social consideration	Make socializing easier	
	More acceptable than smoking ordinary cigarettes to people around me.	
	People in the media or other public figures use HTPs	
	Experts like doctors and scientists use HTPs	
	Family or friends use HTPs	
Product attractiveness	Taste good	
	The packaging is attractive	
	The heating/charging device is attractive	
Personal benefits	I save money by using HTPs instead of smoking ordinary cigarettes	
	I enjoy using HTPs	
	Control my appetite and/or weight	
	Reduce my stress	
	Makes me look cool	
	Give me something to do, to occupy my time	
Help cut down cigarette smoking	Using them helps me cut down on the number of cigarettes I smoke	Current smokers only
Help quit smoking	Using them might help me stop smoking cigarettes	
Help stay somking	Replacing some of my ordinary cigarettes with heated tobacco products means I don’t have to give up smoking cigarettes altogether	

**Table 2 ijerph-17-08030-t002:** Characteristics of current and former smokers using HTPs.

Characteristic	Overall		Current Smokers		Former Smokers	
	*n*	%	*n*	%	*N*	%
Gender						
Male	419	63.7	354	64.5	65	59.6
Female	239	36.3	195	35.5	44	40.4
Age Group						
20–29	126	19.2	116	21.1	10	9.2
30–39	206	31.3	175	31.9	31	28.4
40+	326	49.5	258	47.0	68	62.4
Mean Age (SD)	41.4 (11.6)		41.0 (11.8)		43.3 (10.0)	
Tobacco Use Status						
Current Smoker	549	83.4	549	100.0	0	0.0
Former Smoker	109	16.6	0	0.0	109	100.0
Smoking Status						
Daily	524	79.6	524	95.4	0	0.0
Non-daily *	25	3.8	25	4.6	0	0.0
Former Smokers	109	16.6	0	0.0	109	100.0

* Smoke at least monthly but less than daily in the past 30 days; SD: standard deviation.

**Table 3 ijerph-17-08030-t003:** Reasons for using HTPs among current and former smokers in order of prevalence.

Category/Reason	Overall	Current Smokers	Former Smokers	Current vs. Former Smokers (*p*-Value)
	Weighted Percentage (95% Confidence Interval)			
**Harm reduction—less harmful to themselves than smoking**	**90.6 (87.7–92.9)**	**88.0 (84.5–90.7)**	**95.8 (89.1–98.4)**	**<0.01**
**Harm reduction—less harmful to others**	86.7 (83.2–89.5)	**83.8 (79.9–87.2)**	**91.4 (84.2–95.4)**	**<0.05**
Personal benefits—enjoyment	76.5 (71.9–80.5)	75.2 (70.8–79.2)	78.8 (67.3–87.0)	n.s.
Social consideration - more acceptable to others	74.4 (69.4–78.9)	72.5 (67.5–77.1)	77.7 (66.1–86.2)	n.s.
**Personal benefits—stress reduction**	61.8 (56.6–66.7)	**52.8 (47.6–57.9)**	**76.7 (66.3–84.6)**	**<0.001**
Social influence—family or friends use HTPs	55.9 (50.6–61.1)	58.9 (54.1–63.6)	50.9 (39.3–62.3)	n.s.
Personal benefits—give me something to do, to occupy my time	51.8 (46.4–57.1)	54.1 (49.0–59.1)	47.9 (36.7–59.4)	n.s.
**Personal benefits—taste good**	45.1 (39.8–50.5)	**37.5 (32.9–42.3)**	**58.6 (47.1–69.3)**	**<0.001**
**Convenience—use in places where smoking cigarettes is banned**	44.5 (39.3–49.9)	**49.4 (44.5–54.4)**	**36.2 (25.2–48.2)**	**<0.05**
**Personal benefits—makes socializing easier**	35.4 (30.4–40.7)	**40.0 (35.0–45.2)**	**27.3 (17.8–39.5)**	**<0.05**
Social influence—people in the media or other public figures use HTPs	29.0 (24.0–34.6)	31.3 (26.3–36.7)	25.4 (16.2–37.5)	n.s.
**Product attractiveness—the heating/charging device is attractive**	27.7 (23.4–34.5)	**33.4 (28.8–38.4)**	**17.5 (10.1–28.6)**	**<0.01**
Personal benefits—save money	20.1 (16.1–24.8)	20.3 (16.5–24.7)	19.7 (11.5–31.5)	n.s.
**Product attractiveness—attractive packaging**	16.1 (13.0–19.8)	**22.3 (18.2–27.0)**	**4.0 (1.2–12.3)**	**<0.001**
Personal benefits—control appetite and/or weight	16.2 (12.7–20.4)	19.3 (15.5–23.7)	10.3 (4.3–22.9)	n.s.
**Personal benefits—look cool**	12.5 (9.7–15.8)	**15.9 (12.5–20.0)**	**5.3 (1.9–14.0)**	**<0.001**
Social influence—experts like doctors and scientists use HTPs	10.3 (7.4–14.1)	12.3 (9.0–12.7)	6.8 (2.9–15.2)	n.s.
Reduce smoking—using HTPs helps me cut down on the number of cigarettes I smoke	NA	64.4 (59.5–69.0)	NA	NA
Help quit smoking—using HTPs might help me quit smoking cigarettes	NA	55.1 (49.9–60.2)	NA	NA
Help stay smoking—replacing some of my cigarettes with a heated tobacco product means I don’t have to give up smoking cigarettes altogether	NA	52.0 (46.7–57.2)	NA	NA

Bolded cells are those where the difference between current and former smokers is statistically significant at 0.05 level or lower; n.s.: not statistically significant.

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
