# Peer review of "Reasons for Regularly Using Heated Tobacco Products among Adult Current and Former Smokers in Japan: Finding from 2018 ITC Japan Survey"

_ijerph, 2020, doi:10.3390/ijerph17218030_

Round 1

Reviewer 1 Report

This research is well performed and adds interesting knowledge about dual use of heat-not-burn tobacco products and cigarettes in Japan. Given the interest in this subject, I expect that the paper will be useful to researchers in public health and policy.

My comments are the following.

  1. “To our knowledge, this is the first industry-independent population study on reasons for regularly using HTPs”. This may be true, as long as the terms are defined narrowly enough, but it misleads readers into remaining ignorant of other similar studies based on surveys, not connected with industry, and asking about reasons for use of HTP, for example the study mentioned later in the paper: Kim J, Yu H, Lee S, et al. Tob Control 2018;27:s74–s77. What exactly is the claim to novelty?
  2. “This cross-sectional study comprised of 549 concurrent users (smokers who also use an HTP at 96 least weekly), and 109 former smokers (quit smoking cigarettes completely but use an HTP at least 97 weekly).”
  3. “Referring to the Patel, D. et. al’s 108 category of reasons for current e-cigarette use…” is awkward phrasing.
  4. “to help don’t have to give up smoking altogether” is awkward phrasing. Unless this is the literature translation of the survey question, I suggest rephrasing as “to lessen the need to give up smoking altogether” or similar. This appears to be stated much more clearly already in the final row of Table 3.
  5. “Models were conducted by incorporating complex survey sample designs, including stratification and rescaled cross-sectional weights for statistical inferences to minimize potential systematic errors caused by unequal selection probabilities and ensure that results were nationally representative.” Unless population or subpopulation totals are being estimated, pure rescaling of the cross-sectional weights does NOT affect estimation or inference. (Of course, accounting for stratification and weighting themselves do matter.) The authors should clarify what they mean by rescaling and what it would affect, if anything.
  6. “The predicted marginal standardization method in the SUDAAN (version 11) GEE model (PREDMARG) was used for estimating prevalence. General linear contrasts of the predicted marginals in the corresponding models were specified for significance testing of percent differences between current smokers and former smokers.” I think that this text can be made more specific, since GEE is a framework that incorporates many possible specifications of the mean. I assume that the logistic form was used, but this should be made explicit, since it isn’t the only choice for binary data.
  7. It does not seem that the text at the top of page 7 should be in italics.

Reviewer 2 Report

Broad comment:

This is an interesting study, which aimed to examine reasons for HTP use among a large national sample of Japanese smokers and former smokers. Taking into account the high market growth of heated tobacco products (HTPs) in several countries, which seemed to have been introduced “silently” and without adequate scientific evidence regarding the health impact of their use, it is of high importance for public health to study the determinants of use of this tobacco product.

Specific comments:

Introduction

  • The Introduction provides enough background and an extensive justification for the hypothesis presented in the study. The references are adequate.
  • Lines 77-82: This sentence is too long and should be reformulated in order to be clearer.

Methods

  • The research design is appropriate, and the methods are adequate and very well described.

Results

  • In general, the results are clearly presented.

Discussion and conclusions

  • The results are well discussed.
  • Table 2 - please present standard deviations for the means of the continuous variables; homogenise in the table the number of decimal places for “zero”.
  • Correct several typos along the manuscript, as the frequencies in lines 247, 249 and 260, for example (e.g. 3.5%).

Reviewer 3 Report

This manuscript has addressed an emerging and important public health phenomenon among a unique set of population. The findings have good implications to the research and policy communities. The writing is mostly clear and well-organized. Below I listed my main questions and suggestions.  

  1. Some acronyms need to be spelled out given the international nature of this journal’s audience. For instance, “FDA” in line 44.
  2. Space should be added between 6 and broad (line110).
  3. “For analysis purposes, “refused” or “do not know” responses were excluded” (line 118) On average, what is the invalid response rates in the survey? Which question has the highest invalid response rates?
  4. “The model was not controlled for education and income because they were not significant in the initial bivariate analyses”. (line 130) I’m kind of surprised to see that education and income are not significant to the outcome variables. Is this due to the sample characteristics? Is it the same case in prior research studying cigarette or e-cigarette use?
  5. Line 173-179, is it necessary to put those information in italics?
  6. In the discussion section, the authors related some of the key findings to the broader Japanese culture. As a culture that is known to be relatively more gendered compared to other high-income countries, have the authors found any significant variations in HTP use patterns and the reasons for HTP use by gender? Is gender significantly associated with some of the items on reasons for HTP use?
  7. Based on those significant differences by group, it seems that the current smokers are more likely to be motivated by external, social factors for HTP use, i.e., product attractiveness and making socializing easier. By contrast, the former smokers are more likely to be drawn to something more internal and self-oriented, i.e., less health harm and stress reduction. Why is that? What are the possible implications for future marketing strategies and public health policies? The authors may want to briefly discuss this pattern of group variation in the discussion section.
  8. Rewording is needed: “they were also less likely to reporting using HTPs to cut down on smoking (64.4%) concurrent NVP-cigarette users…” (line 266-267)

Reviewer 4 Report

1, According to the authors, the objectives of the present study to examine reasons for HTP use among a large national sample of Japanese smokers and former smokers, but it is not clear that how the authors support the objectives in the analysis. Or in other words, what is the research question or the what are the research hypothesis ot this study? 2, Please clarify the dependent variables in the method section. If the objectives of the present study is to examine reasons for "HTP use", the key dependent variable of the analysis model should be variables about "HTP use" in general. 3, In general, the price of "HTP use" and cigarette should be added in the analysis. Other information, such as SES and family characteristics (number of smoker in the family, the number of children living in the family, etc.). Number of smoking frienders and other observable confounders should be added in the analysis too.

Round 2

Reviewer 4 Report

The authors have already tried to address my comments.